# TRPM2 Non-Selective Cation Channels in Liver Injury Mediated by Reactive Oxygen Species

**DOI:** 10.3390/antiox10081243

**Published:** 2021-08-03

**Authors:** Eunus S. Ali, Grigori Y. Rychkov, Greg J. Barritt

**Affiliations:** 1Department of Medical Biochemistry, College of Medicine and Public Health, Flinders University G. P. O. Box 2100, Adelaide, SA 5001, Australia; eunus.ali@northwestern.edu; 2School of Medicine, The University of Adelaide, and South Australian Health and Medical Research Institute, Adelaide, SA 5005, Australia; grigori.rychkov@adelaide.edu.au

**Keywords:** liver, reactive oxygen species, TRPM2 channels, Ca^2+^, acetaminophen, ischemia–reperfusion, non-alcoholic fatty liver disease, curcumin

## Abstract

TRPM2 channels admit Ca^2+^ and Na^+^ across the plasma membrane and release Ca^2+^ and Zn^2+^ from lysosomes. Channel activation is initiated by reactive oxygen species (ROS), leading to a subsequent increase in ADP-ribose and the binding of ADP-ribose to an allosteric site in the cytosolic NUDT9 homology domain. In many animal cell types, Ca^2+^ entry via TRPM2 channels mediates ROS-initiated cell injury and death. The aim of this review is to summarise the current knowledge of the roles of TRPM2 and Ca^2+^ in the initiation and progression of chronic liver diseases and acute liver injury. Studies to date provide evidence that TRPM2-mediated Ca^2+^ entry contributes to drug-induced liver toxicity, ischemia–reperfusion injury, and the progression of non-alcoholic fatty liver disease to cirrhosis, fibrosis, and hepatocellular carcinoma. Of particular current interest are the steps involved in the activation of TRPM2 in hepatocytes following an increase in ROS, the downstream pathways activated by the resultant increase in intracellular Ca^2+^, and the chronology of these events. An apparent contradiction exists between these roles of TRPM2 and the role identified for ROS-activated TRPM2 in heart muscle and in some other cell types in promoting Ca^2+^-activated mitochondrial ATP synthesis and cell survival. Inhibition of TRPM2 by curcumin and other “natural” compounds offers an attractive strategy for inhibiting ROS-induced liver cell injury. In conclusion, while it has been established that ROS-initiated activation of TRPM2 contributes to both acute and chronic liver injury, considerable further research is needed to elucidate the mechanisms involved, and the conditions under which pharmacological inhibition of TRPM2 can be an effective clinical strategy to reduce ROS-initiated liver injury.

## 1. Introduction

The liver is central to whole body metabolism and to the co-ordination of metabolic pathways for carbohydrate, lipid, protein, and amino acid metabolism (reviewed in [1]). In addition, the liver has a number of specialised functions, including the metabolic conversion and secretion of drugs and other xenobiotic compounds, and the synthesis and secretion of bile acids [1]. Most of these metabolic reactions occur in hepatocytes (liver parenchymal cells), which constitute about 60 to 80% of the total liver mass (reviewed in [1]). Several other cell types are also important for liver function. These are cholangiocytes, fibroblasts, stellate cells, Kupffer cells (liver resident macrophages), and sinusoidal endothelial cells [1,2]. 

Many acute and chronic liver diseases are recognised. For almost all of these, oxidative stress and reactive oxygen species (ROS) play a significant role in the initiation and progression of the disease [1,3,4,5,6,7,8,9]. Those liver diseases, or liver injuries, especially important when considering the role of oxidative stress, are acute liver injury caused by drug toxicity or ischemia–reperfusion injury, and the chronic liver diseases hepatitis B and C, alcoholic liver disease, non-alcoholic fatty liver disease (NAFLD) and non-alcoholic steatohepatitis (NASH). If untreated, these chronic diseases can lead to fibrosis, cirrhosis, and hepatocellular carcinoma. ROS also are important for this progression [1,10]. Hepatocellular carcinoma is generally not detected until at a late stage, when it is difficult to treat (reviewed in [1,10]). For the treatment of advanced liver disease there are currently few pharmacological options. Surgical liver resection or liver transplant offers the best current treatment (reviewed in [1,11]).

The ROS involved in inducing and promoting the liver diseases listed above are summarised in Figure 1. This also shows the organelles and pathways that can generate superoxide, the precursor of most other ROS [7,12]. The initial step in generation of ROS involves the formation of superoxide from molecular oxygen (O_2_). The chronic diseases or acute injuries listed above provide intracellular conditions favourable for increased activity of the reactions that generate superoxide and/or a rapid increase in O_2_, such as what occurs in ischemia–reperfusion [1,3,4,8,10]. H_2_O_2_, which is formed in vivo from superoxide (Figure 1), is reasonably stable in solution in the laboratory, and has been widely used experimentally as a source of ROS in in vitro studies of liver oxidative stress employing liver cells in culture. In addition to its detrimental role in contributing to ROS-induced liver injury, H_2_O_2_ in vivo initiates some normal intracellular signalling pathways. Also shown in Figure 1 are the reactive nitrogen species that can be formed from superoxide. Several enzymes are involved in the removal of ROS in the liver. Of most importance are glutathione peroxidase, the peroxiredoxin family, and catalase [7,12].

Many intracellular signalling pathways are involved in mediating ROS-induced injury to liver cells. Arguably, one of the most central and important primary pathway is the ROS-induced increase in Ca^2+^ entry and the subsequent increase in the cytoplasmic Ca^2+^ concentration ([Ca^2+^]_cyt_) [1,10,13,14]. This, in turn, leads to increases in organelle Ca^2+^ concentrations and the activation of many downstream intracellular signalling pathways. So far, this Ca^2+^-mediated pathway has been somewhat underemphasised in the thinking about ROS-initiated liver injury and disease. 

Several Ca^2+^-permeable channels in the plasma membrane are known to be activated by ROS [15,16,17,18]. On the basis of studies to date, one of the most important of these is the transient receptor potential melastatin (TRPM2) non-selective cation channel, which facilitates the entry of Ca^2+^ and Na^+^ [19,20,21]. TRPM2 channels have been shown to play important roles in mediating cell death and injury in many organs and cell types [22,23,24,25,26,27,28]. These include paracetamol-induced liver toxicity and liver ischemia and reperfusion injury [15,29,30,31,32].

The aim of this review is to summarise the current knowledge of the roles of TRPM2 channels and intracellular Ca^2+^ in mediating ROS-initiated liver injury and the progression of liver disease and liver injury to end-stage liver disease or liver failure. Of particular interest is the role of TRPM2 in liver injury induced by paracetamol toxicity and ischemia–reperfusion, and in the progression of non-alcoholic fatty liver disease to fibrosis, cirrhosis, and hepatocellular carcinoma. Based on the results of studies conducted so far, TRPM2 channels may offer a suitable pharmacological target for the prevention and/or management of ROS-induced liver disease. Therefore, the final section of the review evaluates currently known TRPM2 inhibitors, including the natural product curcumin. The focus of this review is on the TRPM2 channels expressed in hepatocytes and in other cell types present in liver. However, it is important to note that the TRPM2 channels in some other organs, such as the pancreas and brain, and in white blood cells, also play important indirect roles in the progression of ROS-mediated liver diseases, such as diabetes [33,34,35]. However, we have not included TRPM2 channels in other organs or blood cells in this review. Before reviewing studies on the role of TRPM2 in mediating liver damage in specific pathologies, it is useful to briefly summarise the current knowledge of the expression and location of TRPM2 proteins in animal cells, and of the TRPM2 structure and the mechanism of ROS-initiated TRPM2 activation. 

## 2. TRPM2 Non-Selective Cation Channels Are Formed from Tetramers of the TRPM2 Polypeptide Principally in the Plasma Membrane and Lysosomes, and Are Activated by Allosteric Binding of ADP-Ribose 

TRPM2 channels in animal cells are principally located in the plasma membrane where, when activated, they exert their predominant effect on intracellular Ca^2+^ signalling and downstream signalling pathways [19,20]. However, as discussed below, in some animal cells functional TRPM2 channels are also located in lysosomes, and likely also in mitochondria and possibly the nucleus. Functional TRPM2 channels are formed from homo-tetramers of the TRPM2 polypeptide, which is shown schematically in Figure 2. This depicts a single TRPM2 polypeptide located in the plasma membrane. Each TRPM2 polypeptide (TRPM2 monomer) consists of a cytoplasmic N-terminus (minimal TRPM-homology region (MHR)), six membrane-spanning regions, and a cytoplasmic C-terminus. The C-terminus harbours the TRP helices, H1 and H2, and the NUDT9-homology (NUDT9H) domain (Nudix motif) [19,20,36,37]. The transmembrane spanning domain contains helices S1–S4, which comprise a voltage sensor-like domain, and helices S5 and S6, which comprise the channel pore. Several splice variants of human TRPM2 are known [19]. These include deletions in the N-terminal TRPM homology domain and in the C-terminal NUDT9H domain, as well as a deletion of all the C-terminal regions starting from transmembrane helix S3.

The activation of the TRPM2 channels requires the binding of ADP-ribose and Ca^2+^ to the NUDT9H domain and the interaction of the TRPM2 polypeptide with phosphatidylinositol-4,5-bisphosphate in the membrane (shown schematically in Figure 2) [19,20,36,37]. Phosphatidylinositol-4,5-bisphosphate is thought to interact with basic amino acids in the “membrane interfacial cavity” of the TRPM2 polypeptide [38]. This cavity includes amino acid sequences in the N-terminal MHR4 and pre-S1 domains, and in the junction between the S4 and S5 transmembrane domains and in the TRP domain. Interaction of phosphatidylinositol-4,5-bisphosphate with these basic amino acids is thought to facilitate the allosteric activation by ADP-ribose and Ca^2+^. Under conditions of oxidative stress, the synthesis of intracellular ADP-ribose is enhanced. The two main pathways for ADP-ribose synthesis are from cellular NAD, catalysed by poly(ADP ribose) polymerase (PARP) and poly(ADP ribose) glycohydrolase (PARG) located in the nucleus, and from mitochondrial NAD, catalysed by mitochondrial NAD-ase (Figure 2). Experimentally, and likely also in cells subject to oxidative stress, TRPM2 can be activated by H_2_O_2_. There is some evidence that activation of TRPM2 by H_2_O_2_ is independent of the generation of ADP-ribose and the binding of ADP-ribose to TRPM2. However, it is most likely that the mechanism by which H_2_O_2_ activates TRPM2 involves the stimulation of intracellular ADP-ribose synthesis and interaction of ADP-ribose with TRPM2 as just described. Possibly, also, H_2_O_2_ may act synergistically with ADP-ribose at the ADP-ribose binding site in the NUDT9H domain (reviewed in [19]). 

While recent studies of TRPM2 structure employing X-ray crystallography and cryo-electron microscopy have uncovered further details of the mechanisms by which ADP-ribose and Ca^2+^ induce conformational changes in TRPM2 polypeptides, the molecular interactions involved are still not yet fully understood [36,39,40,41]. Moreover, there are subtle differences in the molecular interactions and conformational changes involved in the activation of TRPM2 in different species [41]. Separate from these considerations, in some cell types, including hepatocytes (discussed below), activation of TRPM2 channels at the plasma membrane also involves trafficking of the TRPM2 polypeptides from an intracellular location to the plasma membrane [42,43].

TRPM2 channels are non-selective cation channels that, in the plasma membrane, principally admit Ca^2+^ and Na^+^ to enter the cytoplasmic space down a concentration gradient [19,21,44]. The permeability to Ca^2+^ is about equal to that of Na^+^ [21,44]. As mentioned below, it has been suggested that in lysosomes and mitochondria, Zn^2+^ may also be translocated through TRPM2 channels [45,46,47,48]. The requirement for Ca^2+^ in the activation pathway for TRPM2 creates a positive feedback loop wherein an increase in Ca^2+^ at the intracellular mouth of the channel enhances further activation [36]. Moreover, in cell types that express active (Na^+^-Ca^2+^) exchange proteins in the plasma membrane, an increase in Na^+^ in the cytoplasmic space at the mouth of TRPM2 can also activate further Ca^2+^ entry driven by the (Na^+^-Ca^2+^) exchanger [19,49].

TRPM2 proteins have been detected in lysosomes in several cell types [50,51,52,53,54]. These include pancreatic ß-cells, neuronal cells, dendritic cells, macrophages, and vascular smooth muscle cells. TRPM2 proteins have also been detected in mitochondria in neuronal cells [45,46]. Other studies with prostate, oral squamous, and breast cancer cells have provided evidence that, in these cancer cells, some TRPM2 protein is present in the nucleus [55,56,57]. Nuclear location of the TRPM2 protein was not detected in non-cancerous cells. The authors suggest that TRPM2 located in the nucleus may be involved in the regulation of cell proliferation in cancer cells.

There is evidence that TRPM2 proteins in lysosomal membranes form functional TRPM2 channels in [50,52,54]. However, to our knowledge, functional TRPM2 channels have not yet been identified in the mitochondria or in any other intracellular organelle. In lysosomes, activated TRPM2 channels are thought to release Ca^2+^ and Zn^2+^ into the cytoplasmic space [45,46,47,48]. In pancreatic ß-cells, the subsequent increase in [Ca^2+^]_cyt_ is proposed to amplify the physiological intracellular Ca^2+^ signals [52]. In vascular smooth muscle cells and in macrophages, Ca^2+^ release via TRPM2 channels is proposed to increase lysosomal acidification, leading to enhanced autosomal degradation. This, in turn, is proposed to lead to cell death in smooth muscle cells and to be responsible for bactericidal activity in macrophages [50,51]. 

It has been suggested that the release of Zn^2+^ from lysosomes via TRPM2 channels may play a role in the “delayed” neuronal cell death that follows ischemia and reperfusion in the brain, such as in stroke [45,46,47,48]. On the basis of experiments conducted with mouse hippocampal neurons and neuroblastoma SH-SY5Y cells, employing H_2_O_2_ as a source of ROS, Li and colleagues have proposed that the Zn^2+^ released from lysosomes in response to the activation of lysosomal TRPM2, initiated by ROS, enters the mitochondria via mitochondrial TRPM2 channels, leading to an exacerbation of mitochondrial ROS production. The authors propose that, as a result, and over time, the neuronal cells die. They suggest that this mechanism may underlie “delayed” neuronal cell death following brain ischemia [45,46,47,48]. However, other studies have shown that Zn^2+^ inactivates TRPM2 as well as some other ion channels, possibly by covalent modification [58,59]. Moreover, Zn^2+^ also has the capacity to reduce ROS through non-enzymatic reactions [60]. Further studies are needed to confirm the location of the TRPM2 protein and the presence of functional TRPM2 channels in mitochondria, and to investigate the possible movement of Zn^2+^ through TRPM2. 

## 3. TRPM2 Protein and Functional TRPM2 Channels Are Expressed in Liver Cells

In human liver, expression of TRPM2 mRNA has been detected in fixed tissue by in situ hybridisation in hepatocytes and Kupffer cells [15]. Expression of TRPM2 mRNA and protein in liver tissue and in isolated mouse and rat hepatocytes has been detected using qPCR, Western blot, and immunofluorescence [15,29,32,43]. Functional TRPM2 channels in the plasma membranes of isolated mouse and rat hepatocytes have been characterised using whole cell patch clamp recording, fluorescence imaging of [Ca^2+^]_cyt_, and pharmacological inhibitors (Figure 3) [29]. Thus, it can presently be concluded that functional TRPM2 channels are present in hepatocytes. To assess whether functional channels are present in Kupffer cells, and in other cell types in the liver, will require further experiments.

## 4. TRPM2 Channels Mediate Liver Injury Induced by Acetaminophen (Paracetamol) Toxicity

Acetaminophen (paracetamol) is widely used to treat many types of pain, and if used at the recommended dose has few side effects [61,62]. However, acetaminophen hepatotoxicity due to acetaminophen overdose is a leading cause of acute liver toxicity [61,63,64]. Recommended therapeutic doses of acetaminophen are up to a maximum of about 55 mg/kg body weight per day, which yields approximately 5–10 mg/L (0.3–0.7 mM) acetaminophen in the vasculature of the liver and in target organs. By comparison, toxic doses are greater than about 125 mg/kg body weight, often taken as a single dose. This can yield a concentration of acetaminophen of about 200–300 mg/L (1.3–2 mM) in the vasculature of the liver and other organs [61,65,66,67]. Once initiated, acetaminophen hepatotoxicity is difficult to reverse and if untreated can lead to liver failure [61,64]. Administration of the antioxidant N-acetyl cysteine is currently widely used to treat patients suffering from acetaminophen-initiated liver toxicity [61,64]. However, to be effective, N-acetyl cysteine needs to be administered very soon after the acetaminophen overdose. Alternatives to N-acetyl cysteine for the treatment of acetaminophen-initiated liver toxicity would be most valuable to improve patient outcomes [26,61,64].

Acetaminophen is initially metabolised in hepatocytes by three main enzymes: UDP-glucuronosyl transferase, sulphotransferase, and cytochrome P450 2E1, as shown schematically in Figure 4 [64,68]. The products of the UDP-glucuronosyl transferase and sulphotransferase reactions, acetaminophen glucuronate and acetaminophen sulphate, respectively, are eliminated by excretion in bile fluid. At low (“normal”) doses of acetaminophen, the product of cytochrome P450 2E1, N-acetyl-p-benzo-quinone imine (NAPQI), is converted to mercapturic acid in a reaction with glutathione, catalysed by glutathione S-transferase, and the product, mercapturic acid, is also eliminated by excretion in bile fluid [64,68]. However, at high levels of acetaminophen, as in acetaminophen toxicity, glutathione S-transferase becomes fully saturated, glutathione is depleted, and NAPQI accumulates [64,68]. Under these conditions, NAPQI will react irreversibly with cysteine groups in enzymes and other proteins, and with DNA, leading to protein inactivation and damaged DNA. Moreover, the depletion of glutathione impairs the capacity of hepatocytes to remove ROS, generated as a result of liver injury induced by NAPQI [64,68]. This combination of events leads to mitochondrial dysfunction, decreased ATP production, and increases in ROS and in [Ca^2+^]_cyt_, culminating in necrotic cell death [15,26,29,30,61,64,68,69].

Although the pathways are not yet fully understood, the increase in [Ca^2+^]_cyt_ initiated by acetaminophen overdose appears to play a central role in the initiation and progression of necrosis [26,29]. Several studies have investigated the role of Ca^2+^ entry to hepatocytes through TRPM2 channels in mediating acetaminophen-induced hepatocyte death [15,29,30].

Kheradpezhouh and colleagues investigated the ability of acetaminophen to induce Ca^2+^ entry to hepatocytes isolated from the livers of mice and rats [29]. Whole cell patch clamp recording and Ca^2+^ imaging were used to show that, in isolated hepatocytes in culture, acetaminophen increases Ca^2+^ entry through channels with the same biophysical and pharmacological characteristics as those activated by H_2_O_2_ or intracellular ADP-ribose (Figure 3) [29]. Ca^2+^ entry initiated by acetaminophen or H_2_O_2_ was not observed in hepatocytes isolated from TRPM2 knockout mice, compared with hepatocytes isolated from wild-type mice. The characteristics of the channels activated by acetaminophen were the same as those reported for TRPM2 in other cell types [29]. Knockdown of TRPM2, using siRNA, in wild-type mouse hepatocytes ablated the activation of channels with the characteristic of TRPM2 by acetaminophen or H_2_O_2_. In TRPM2 knockout mice, liver injury (assessed by measurement of blood liver marker enzymes and liver histology) induced by acetaminophen was substantially reduced compared with that in wild-type mice. It was concluded that Ca^2+^ entry to hepatocytes via TRPM2 channels, initiated by ROS and mediated by ADP-ribose, makes a significant contribution to acetaminophen-induced hepatocyte death [29]. 

Studies of the mechanism by which acetaminophen activates TRPM2 channels in hepatocytes showed that TRPM2 channel activation, initiated by acetaminophen or H_2_O_2_, is relatively slow in onset [43]. The time required to achieve maximal current development was about 210 s. In HEK293T cells heterologously expressing TRPM2, the time required to achieve maximal current development was about 80 s. Immunofluorescence, antibodies specific for TRPM2 and cadherin (plasma membrane marker), and confocal microscopy were employed to study the intracellular location of TRPM2 in rat hepatocytes [43]. In hepatocytes cultured under normal conditions, considerable TRPM2 was found in the cell interior (Figure 5, TRPM2 panel). Treatment with H_2_O_2_ or acetaminophen induced co-localisation of TRPM2 and cadherin at the plasma membrane (Figure 5, Merged and Zoom area panels). Movement of TRPM2 to the plasma membrane was confirmed using sulfo-NHS-SS-biotin to label the proteins on the cell surface. The increase in TRPM2 protein at the plasma membrane was associated with an increase in TRPM2 channel activity. Taken together, these results provide evidence that trafficking of TRPM2 from an intracellular location, possibly endosomes, to the plasma membrane contributes to acetaminophen-initiated activation of TRPM2 in hepatocytes [43].

Experiments conducted by Wang and colleagues have also provided evidence of the role of TRPM2 in acetaminophen-initiated hepatocyte injury [30]. These authors showed that, in isolated mouse hepatocytes, acetaminophen initiates an increase in ROS and activates Ca^2+^ entry, which was inhibited by clotrimazole, an inhibitor of TRPM2. Experiments employing HeLa cells heterologously expressing TRPM2 led to the proposal that Ca^2+^ entry via TRPM2 activates Ca^2+^/calmodulin-dependent protein kinase II, which, in turn, phosphorylates Beclin 1, supresses autophagy, and increases the susceptibility of hepatocytes to cell death [30]. (Beclin 1 forms a complex with class III phosphatidylinositol 3-kinase and several other proteins involved in the initial steps in the assembly of autophagosomes [30]).

TRP channels other than TRPM2 may also be involved in acetaminophen-initiated Ca^2+^ entry into hepatocytes. Badr and colleagues investigated the roles of several TRP channels known to be activated by ROS in mediating acetaminophen-initiated increases in [Ca^2+^]_cyt_ in immortalised human HepG2 liver cells [15]. The channels investigated were TRPVs 1, 3, and 4; TRPCs 1, 4, and 5; TRPMs 2 and 7; and TRPA1. Measurement of channel expression using RT-PCR indicated that TRPV1, TRPV3, TRPC1, and TRPM7 are expressed at the highest levels in HepG2 cells. Ca^2+^ imaging was employed to assess Ca^2+^ entry via increases in [Ca^2+^]_cyt_, and siRNA directed against a given TRP channel and pharmacological inhibitors were used to identify the channels activated by acetaminophen. It was concluded that treatment of HepG2 cells with acetaminophen or H_2_O_2_ activates Ca^2+^ entry through the TRPV1, TRPC1, TRPM2, and TRPM7 channels, with TRPV1 and TRPC1 being the major contributors [15]. HepG2 liver cells are immortalised cells originally derived from a human liver tumour and hence differ substantially from hepatocytes in situ in their morphology, expression of proteins, and other properties (reviewed in [1]). Therefore, it will be important to conduct further experiments with animal models of acetaminophen toxicity to fully evaluate the potential roles of these other TRP proteins.

Another study by Echtermeyer and colleagues suggests that TRPV4 may also contribute to acetaminophen-initiated hepatocyte toxicity [69]. Liver damage, assessed by measurement of blood liver marker enzymes and liver histology, was substantially reduced in TRPV4 knockout mice compared to that in wild-type mice, and in wild-type mice treated with the TRPV4 inhibitor HCO67047. Treatment with HCO67047 was compared with treatment with N-acetyl-cysteine and found to be equally effective in reducing acetaminophen-initiated liver injury. TRPV4 knockout or treatment with HCO67047 also reduced acetaminophen-initiated increases in ROS and nitric oxide, and acetaminophen-induced mitochondrial membrane depolarisation. Patch clamp recording with HEK293T cells heterologously expressing TRPV4 showed that NAPQI activates a current with the characteristics of TRPV4. However, such a NAPQI-activated current was not observed in isolated mouse and human hepatocytes [69]. Further experiments are required to confirm that TRPV4 proteins are present in the hepatocyte plasma membrane and do, indeed, form functional TRPV4 channels in freshly isolated hepatocytes.

In conclusion, the results of studies conducted to date provide evidence that Ca^2+^ entry though TRPM2 channels contributes to acetaminophen-initiated hepatocyte injury and death, as shown schematically in Figure 6. Further experiments are needed to define more clearly the sequence and time-course of the events that follow TRPM2 activation and lead to cell death. Moreover, other plasma membrane Ca^2+^-permeable channels, including other TRP family members and store-operated Ca^2+^ channels, may also be involved in mediating acetaminophen (ROS)-initiated Ca^2+^ entry into hepatocytes.

## 5. TRPM2 Channels May Be Involved in Liver Injury Initiated by Ischemia–Reperfusion during Liver Surgery

Surgery, involving liver resection or liver transplant, is presently the best available treatment for advanced cirrhosis, hepatocellular carcinoma, and other advanced liver diseases [11,70,71]. These surgical procedures involve clamping of relevant blood vessels in the liver (ischemia), in order to limit the loss of blood during the resection and insertion of the donor liver [11,70,71]. Following unclamping of the blood vessels, normal blood flow is resumed (reperfusion). However, these surgical procedures can lead to ischemia–reperfusion injury [11,70,71]. This is particularly important in liver transplants in the case of the donor liver, which may have been stored and transported in the ischemic state for some period of time [71,72]. In order to increase the number of available donor livers for liver transplant patients, livers from steatotic and aged donors can potentially be used. However, these livers are more at risk for ischemia–reperfusion injury [11,72]. Ischemia–reperfusion injury also occurs in livers subject to haemorrhagic shock [73]. Thus, strategies to reduce liver ischemia–reperfusion injury would greatly expand the pool of usable donor livers and hence the survival and wellbeing of patients with end-stage liver disease [11,72,74].

The molecular mechanisms involved in liver ischemia–reperfusion injury are complex, multifactorial, and not yet fully understood [11]. The principal cell types involved are the hepatocytes and Kupffer cells (liver resident macrophages) [11]. Ischemia causes a decrease in ATP production, damage to mitochondria, and likely some increase in ROS. Reperfusion associated with re-oxygenation of the affected liver tissue leads to large increases in ROS, and increases in the extramitochondrial and mitochondrial Ca^2+^ concentration [11,13,71,75,76,77,78]. These changes then induce further damage to mitochondria and release of inflammatory cytokines. Some evidence that enhanced Ca^2+^ entry into liver cells and mitochondria play important roles in mediating liver ischemia–reperfusion injury comes from studies with Ca^2+^-channel inhibitors. In mouse and rat liver models of ischemia–reperfusion, pre-treatment with the Ca^2+^-channel blockers 2-aminoethoxydiphenyl borate (2-APB) or Gd^3+^ was found to protect against ischemia–reperfusion injury [13,79,80].

Examples of changes in ROS and Ca^2+^ in livers subject to ischemia–reperfusion are shown in Figure 7, which includes results from several different studies employing mouse and rat in vivo and in vitro models of liver ischemia–reperfusion injury [77,78,81,82]. Notwithstanding differences in the models and experimental procedure, the results indicate the relationships between the increases in ROS, cellular Ca^2+^, and the blood concentrations of alanine transaminase (a marker of liver injury) observed during and after ischemia and reperfusion. In these models, ROS are increased somewhat at 30 min after initiation of re-perfusion and further at 120 min (Figure 7A). These increases in ROS can be correlated with the increases in intracellular Ca^2+^ at 5 and 30 min after initiation of the reperfusion (Figure 7C). Signs of liver injury are evident at 120 min (Figure 7B). Although direct experiments involving the measurement of intracellular calcium in liver cells during ischemia and reperfusion are somewhat limited, no substantial increases in mitochondrial or cytoplasmic Ca^2+^ concentrations in hepatocytes during the ischemic phase have so far been observed [78].

In principle, the biochemical pathways involved in the initiation and progression of ischemia–reperfusion injury in liver are likely similar to those involved in other well-studied examples of organ ischemia–reperfusion injury, including brain, kidney, and heart [22,23,24,25,28]. Ca^2+^ entry via TRPM2 channels in the plasma membrane has been shown to be an important component of the mechanisms responsible for ischemia–reperfusion injury in brain and kidney [22,24,25]. In these tissues, ischemia–reperfusion injury results in an increase in ROS, activation of TRPM2, Ca^2+^ entry, and increased ([Ca^2+^]_cyt_, leading to mitochondrial Ca^2+^ overload, mitochondrial damage, cell injury, and death [22,24,25,81,82,83,84,85,86].

There is also some evidence that TRPM2 channels may contribute to myocardial ischemia–reperfusion injury (reviewed in [22]). It is proposed that during the reperfusion phase Ca^2+^ entry into cardiac myocytes via TRPM2 leads to an increase in Ca^2+^ in the mitochondrial matrix, opening of the mitochondrial transition permeability pore, release of cytochrome C, activation of caspases, and cell death [22]. However, other studies have revealed a role for myocardial TRPM2 in protecting cardiac myocytes from ischemia–reperfusion injury [49,87,88,89]. The initial steps in the proposed protective mechanisms involve activation of TRPM2 by increased ROS, enhanced Ca^2+^ entry, and an increase in [Ca^2+^]_cyt_, as proposed for neuronal and kidney cells subjected to ischemia–reperfusion. However, for cardiac myocytes, the increase in ([Ca^2+^]_cyt_ is proposed to activate and/or increase the expression and of transcription factor HIF1. This induces the synthesis of the antioxidant enzymes superoxide dismutase-1 and-2 (via transcription factor nuclear factor erythroid 2-related factor 2 (Nrf2)), which reduces ROS and protects mitochondria from damage. This, in turn, allows mitochondria to maintain normal ATP synthesis and decreases the production of ROS by mitochondria [89,90]. Further research is needed to clarify these somewhat contradictory roles of TRPM2 in the response of neuronal, kidney, and cardiac myocytes to ischemia–reperfusion. Possibly both the detrimental and beneficial pathways operate in each cell type, with the outcome depending on which pathway predominates.

There have been relatively few studies to date of the role of TRPM2 channels in mediating liver ischemia–reperfusion injury. In one of these, Bilecik and colleagues used a rat model of liver ischemia–reperfusion injury to investigate the effect of ischemia and reperfusion on TRPM2 expression [32]. They observed a very small increase in TRPM2 mRNA expression, measured using qPCR, in livers subject to 60 min ischemia and 60 min reperfusion, compared to sham-operated livers. Small increases in expression of TRPM6, TRPM7, and TRPM8 were also observed. In another study, Li and colleagues used adenovirus interference to knockdown TRPM2 (assessed by qPCR) in a rat model of liver ischemia–reperfusion [31]. Livers in which TRPM2 expression was decreased by about 40% exhibited less injury, assessed by measurement of blood liver marker enzymes and qualitative liver histology, when subjected to ischemia–reperfusion. These changes were associated with increased expression of superoxide dismutase, and reduced myeloperoxidase and ROS, measured as malonyldialdehyde.

The results of the limited studies conducted so far suggest that TRPM2 may mediate Ca^2+^ entry into liver cells in liver ischemia–reperfusion injury. Further systematic investigations are required to test this idea, and to determine whether TRPM2 may have dual roles in liver ischemia–reperfusion: on the one hand mediating cell death and on the other mediating survival of mitochondria and reduction of ROS, as proposed for its role in myocardial muscle cells.

## 6. TRPM2 Channels in Liver Cells May Be Involved in Non-Alcoholic Fatty Liver Disease and Its Progression to Cirrhosis and Hepatocellular Carcinoma

To our knowledge, there have so far been few studies of the role of TRPM2 channels in liver cells in the initiation of liver steatosis and in the progression non-alcoholic fatty liver disease to cirrhosis and hepatocellular carcinoma. Given the important role played by ROS in the development of non-alcoholic fatty liver disease and hepatocellular carcinoma, it is very likely that liver cell TRPM2 channels, activated by ROS, are involved (reviewed in [1,10]). In addition, Ca^2+^ entry via TRPM2 channels in pancreatic ß-cells, neutrophils, and endothelial cells, among other cell types, undoubtably contributes to the onset and progression of liver pathologies associated with type 2 diabetes, hepatitis, and other liver diseases involving ROS and inflammatory pathways [33,34,35,91,92].

There is suggestive evidence that TRPM2 channel activity may be altered in steatotic hepatocytes and hence may contribute to the progression of non-alcoholic fatty liver disease to cirrhosis and hepatocellular carcinoma [93]. Feng and colleagues employed an “in vitro model of non-alcoholic fatty liver disease” comprised of immortalised LO2 liver cells in culture loaded with lipids by incubation with palmitate. Co-incubation of LO2 cells with the antioxidant saliroside (2-(4-hydroxyphenyl)-ethyl-β-D-glucopyranoside), derived from the plant *Rhodiola rosea*) and palmitate reduced lipid accumulation and reduced palmitate-induced cell injury, assessed by measurement of cell viability and the release of alanine transaminase. Lipid loading was associated with an increase in TRPM2 expression, assessed by Western blot, and increases in [Ca^2+^]_cyt_, Ca^2+^/calmodulin-dependent protein kinase II phosphorylation, and expression of mRNAs encoding IL-1beta and IL-6. These palmitate-induced changes were all reduced by saliroside. The authors concluded that the observed actions of saliroside are due to inhibition of expression of TRPM2 and subsequent reduction in [Ca^2+^]_cyt_, leading to decreased activation of Ca^2+^/calmodulin-dependent protein kinase II, increased autophagy, and decreased inflammation [93].

## 7. TRPM2 Channels Are Potential Pharmacological Targets for the Prevention of Liver Injury Induced by Reactive Oxygen Species

As discussed above, Ca^2+^ and Na^+^ entry into hepatocytes and other liver cell types contributes to, or is proposed to contribute to, hepatocyte and liver injury in several ROS-mediated liver pathologies. Pharmacological inhibition of TRPM2 may be a useful strategy to reduce ROS-mediated cell injury to the liver as well as to many other organs [25,26,94,95]. However, efforts to achieve the selective inhibition of TRPM2 channels by agents that can potentially be used in the clinic have been hampered by the absence of TRPM2 inhibitors with the desired pharmacological properties: high affinity and specificity for TRPM2 and good bioavailability [96].

Our current knowledge of TRPM2 inhibitors is summarised in Table 1. The placement of inhibitors into the groups listed in the first column of Table 1 follows the arrangement employed by Zhang and colleagues [96]. Most dose–response curves (IC_50_ values) for a designated inhibitor have been determined for endogenous TRPM2 channels expressed in the cell type under study. These experiments have used patch clamp recording of currents activated by intracellular ADP-ribose and exhibiting the characteristics of TRPM2 currents. Some dose–response curves reported in Table 1 have been determined using HEK293 cells heterologously expressing TRPM2. In these experiments, TRPM2 was activated by addition of H_2_O_2_ or by intracellular ADP-ribose, and Ca^2+^ entry measured by Ca^2+^ fluorescence imaging or patch clamp recording. One of the most commonly used and reasonably specific TRPM2 inhibitors is N-(p-amylcinnamoyl) anthranilic acid (ACA). For inhibitors that are analogues of ADP-ribose, the mechanism of inhibition is reasonably well defined. However, for many other inhibitors, the binding site and mechanism of inhibition are not well understood.

Recent research aimed at improving the specificity of TRPM2 inhibitors has included studies directed towards the synthesis of modifications of ADP-ribose and N-(p-amylcinnamoyl) anthranilic acid in order to find molecules that inhibit TRPM2 at lower concentrations and with greater specificity. Other approaches have involved virtual and high-throughput screening (Table 1). These investigations have resulted in the development of some TRPM2 inhibitors that are more effective than the re-purposed ion channel inhibitors (Table 1). However, none of these newly developed compounds yet possess all the properties suitable for clinical application [96]. These include membrane permeability and bioavailability. One TRPM2 inhibitor listed in Table 1 with a very low IC_50_ value is the natural product curcumin.

## 8. Curcumin, a Natural Product, Has the Potential to Prevent Liver Injury Induced by Reactive Oxygen Species by Inhibiting TRPM2

Curcumin, (1E,6E)-1,7-bis(4-hydroxy-3-methoxyphenyl)-1,6-heptadiene-3,5-dione, is principally derived from the spice turmeric (*Curcuma longa* Linn). Curcumin has been used for centuries to treat liver diseases as well as many other diseases and conditions [120,121,122]. Examples of liver diseases for which curcumin is beneficial include drug-induced liver toxicity, hepatocellular carcinoma, and ischemia–reperfusion injury [122,123,124,125,126]. Many of these beneficial effects of curcumin can be attributed to the antioxidant actions of the curcumin molecule [120,121]. As a result of the long-recorded history of the use of curcumin in the treatment of human disease, and more recent understanding of the biochemistry of this compound, curcumin is classified by the Food and Drug Administration (FDA) of the United States as “generally considered as safe” for human use [127].

The biological actions of curcumin can be principally attributed to three highly reactive functional chemical groups [120,121]. These are the two phenolic moieties and the central diketone moiety (Figure 8). These functional groups, especially the diketone moiety (including the two α-, β-unsaturated carbonyl groups) are responsible for the ability of curcumin to interact with ROS, proteins, DNA, and many metal ions [120,121]. The chemical (non-enzymatic) interaction of curcumin with various species of ROS endows curcumin with powerful antioxidant properties. The chelation of metals by curcumin in non-enzymatic reactions can both remove metals that cause cell toxicity and provide an avenue for curcumin to act as a metal-based antioxidant. The latter is achieved by complexes of curcumin with Cu^2+^ or Mn^2+^ that can interact with superoxide. Non-enzymatic covalent modification of enzymes and other proteins by curcumin, for example covalent modification of cysteine residues, can result in inhibition or activation of the target enzyme [120,121,128,129]. However, the highly reactive functional groups, which provide curcumin with its biological activities, are also responsible for its rapid metabolism by the liver and its low bioavailability when administered to humans and animals [120,121].

Some of the enzymes, ion channels, and other proteins that are covalently modified by curcumin, leading to alterations in the protein structure and enzyme activity, are summarised in Table 2. These are arranged under the metabolic and signalling pathways known to be affected by curcumin (in all cell types and tissues, including the liver). Due to its high chemical reactivity, curcumin can covalently modify several enzymes in a given pathway [129]. Depending on the pathway, the end result may be activation or inhibition. So, the effects of curcumin in the pathways shown in Table 1 are complex. Of particular interest for the present discussion are the actions of curcumin in increasing the expression of the antioxidant enzymes involved in the removal of ROS, and in inhibiting the ion channels, of which TRPM2 is one.

Studies with rat hepatocytes and neuroblastoma cells have provided evidence that curcumin inhibits TRPM2 channels [119,155]. Oz and Celik investigated the actions of curcumin on TRPM2 channels in SH-SY5Y neuroblastoma cells heterologously expressing TRPM2 [155]. They found that pre-incubation with curcumin inhibited TRPM2 currents activated by H_2_O_2_, measured by patch clamp recording.

In rat hepatocytes, curcumin was found to inhibit the activation by ADP-ribose of TRPM2 currents, measured using whole cell patch clamp recording (Figure 9A,B). Curcumin also inhibited Ca^2+^ entry through TRPM2 channels, measured in rat hepatocytes using patch clamp recording and Ca^2+^ imaging, when these were activated by H_2_O_2_ or acetaminophen. The IC_50_ for curcumin, measured in HEK293T cells heterologously expressing TRPM2, was found to be about 50 nM (Figure 9C).

To investigate the mechanism by which curcumin inhibits TRPM2, Kheradpezhouh and colleagues conducted further experiments with HEK293T cells heterologously expressing TRPM2. These experiments employed patch clamp recording, N-acetyl cysteine, addition of curcumin before or after achieving whole cell configuration, washout of curcumin, and dose–response curves at two ADP-ribose concentrations [119]. The results showed that curcumin does not block the TRPM2 channel but prevents the activation of TRPM2 by ADP-ribose. The mechanism does not appear to involve a reduction in ROS and/or an increase in glutathione. Further experiments are required to investigate the possibility that the mechanisms of inhibition of TRPM2 involves covalent modification of a protein involved in the activation pathway or of TRPM2 itself [119].

Taken together, these results provide evidence that curcumin, at concentrations in the nanomolar range, inhibits Ca^2+^ entry through TRPM2 channels in hepatocytes. While details of the molecular mechanisms involved remain to be determined, curcumin may offer an avenue for pharmacological inhibition of TRPM2 channels, and hence inhibition of ROS-initiated liver injury. Curcumin, as a potential pharmacological agent for the clinical treatment of liver disease or injury via inhibition of TRPM2, offers two other advantages. Firstly, in addition to inhibition of Ca^2+^ entry via TRPM2 channels and reduction of the subsequent detrimental consequences of increased [Ca^2+^]_cyt_, curcumin reduces ROS through mechanisms independent of TRPM2 channels, as discussed above [120,121]. Secondly, curcumin has a well-characterised track record of use for treatment of human disease, as also mentioned above [127]. This prior knowledge would reduce the work required for clinical trials needed to validate the pharmacological use of curcumin for the treatment of liver diseases, such as drug toxicity and ischemia–reperfusion injury. Further research is needed to improve the bioavailability of curcumin, although considerable progress has already been made in developing curcumin formulations with enhanced bioavailability that could potentially be taken orally [156,157].

## 9. Conclusions

Functional TRPM2 channels are expressed in the liver in hepatocytes and likely also in Kupffer cells. They mediate Ca^2+^ entry and are activated by increases in ROS and ADP-ribose under conditions of oxidative stress. The resulting increase in [Ca^2+^]_cyt_ plays an important role in causing hepatocyte injury and death in drug-induced liver toxicity, and likely also in ischemia–reperfusion injury and the progression of non-alcoholic steatohepatitis to cirrhosis and hepatocellular carcinoma. Given the importance of ROS in liver diseases and of TRPM2 and intracellular Ca^2+^ in mediating ROS-induced cell injury and death in other cell types and organs, it is somewhat surprising that there have so far been relatively few studies of the role of TRPM2 in oxidative stress in liver diseases. Further research, directed towards better definition of the nature of the downstream pathways and organelles affected by TRPM2-mediated increases in [Ca^2+^]_cyt_, and the time courses involved, will be invaluable. As will further investigation of the balance between the detrimental and beneficial pathways initiated by TRPM2 channels in ROS-mediated liver diseases.

Pharmacological inhibition of TRPM2 offers a strategy to reduce ROS-mediated liver injury, but this approach is currently hampered by lack of a suitable TRPM2 inhibitor with high affinity and specificity that can be delivered clinically with reasonable bioavailability. One candidate for such a potential TRPM2 inhibitor is the natural product curcumin. Apart from its acceptance as being safe for the treatment of human diseases, curcumin has multiple actions, especially non-enzymatic removal of ROS, in addition to inhibition of TRPM2 channels. However, considerable further studies are needed to define the mechanism by which curcumin inhibits TRPM2 and to test its potential clinical usefulness for the treatment of liver diseases.

## Figures and Tables

**Figure 1 antioxidants-10-01243-f001:**
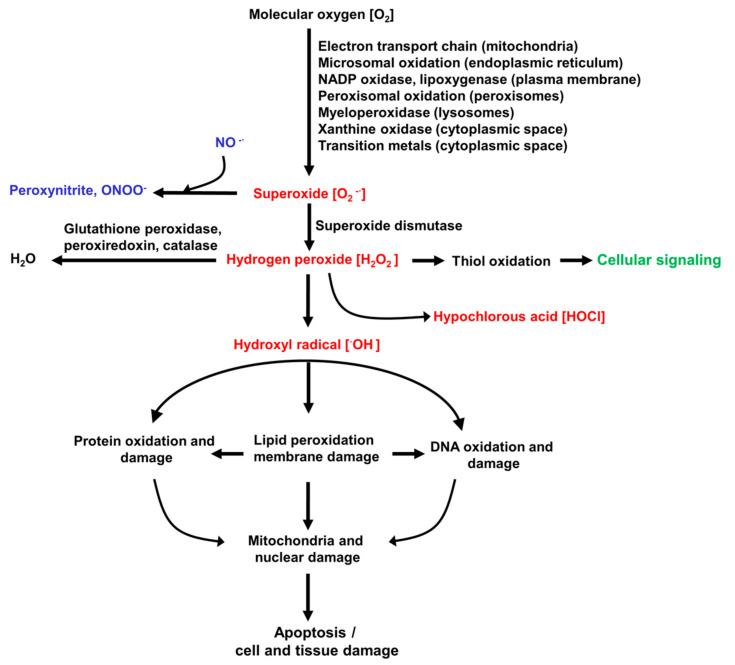
Reactive oxygen species (ROS) involved in liver disease. Shown are the main ROS (in red), the pathways for their formation, and a schematic indication of the downstream damage to proteins, DNA, and organelles that can potentially be caused by ROS. Under normal conditions (non-oxidative stress), H_2_O_2_ also has important intracellular signalling functions (indicated in green). Superoxide can also generate reactive nitrogen species (indicated in blue), which contribute to cellular injury.

**Figure 2 antioxidants-10-01243-f002:**
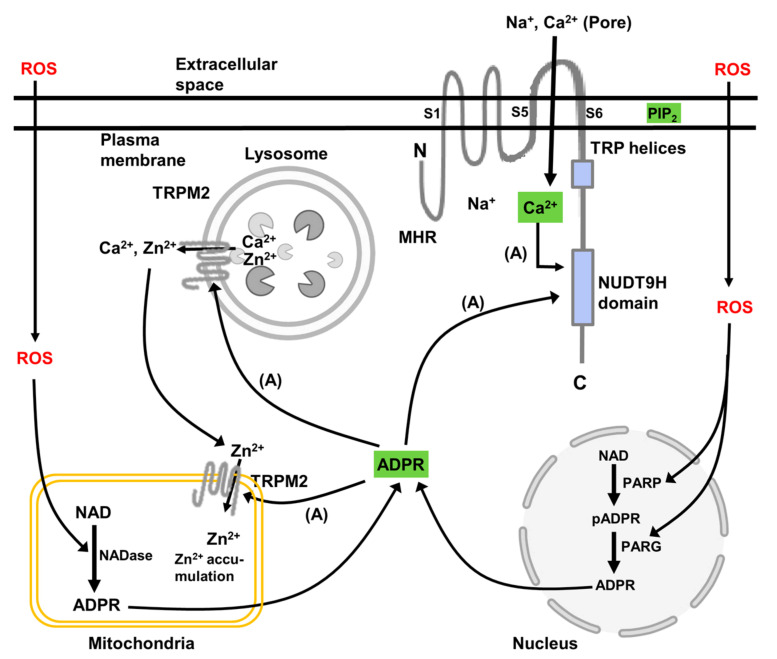
A schematic representation of the intracellular location of TRPM2 channels in animal cells, key functional and regulatory domains in the TRPM2 polypeptide, and the metabolic pathways leading to TRPM2 activation. The scheme shows a single polypeptide of TRPM2 whereas the functional channel is composed of four polypeptides. The domains of TRPM2 indicated are the minimal TRPM-homology region (MHR), the TRP helices, and the NUDT9 homology (NUDT9H) domain (Nudix motif). In animal cells, activation of TRPM2 is initiated by an increase in ROS, which activates poly(ADP-ribose) polymerase (PARP) and poly(ADP-ribose) glycohydrolase (PARG) in the nucleus and NAD-ase in the mitochondria. These enzymes convert NAD to poly(ADP-ribose) (pADPR) and ADP-ribose (ADPR) (shown in green). Together with Ca^2+^ (green) ADP-ribose binds to the NUDT9H domain of TRPM2 and opens the channel. The activation also requires phosphatidyl inositol 4,5 bisphosphate (PIP_2_) (green) in the plasma membrane, which is thought to interact with the basic amino acid sequences in the N-terminal MHR4 and pre-S1 domains, and in the junction between the S4 and S5 transmembrane domains and in the TRP domain. For TRPM2 in the plasma membrane, opening of the channel results in a large entry of Ca^2+^ and Na^+^ into the cytoplasmic space. For TRPM2 in lysosomes and mitochondria, it is proposed that activation of TRPM2 in lysosomes leads to release of Zn^2+^ and Ca^2+^, acidification of the lumen of the lysosome, and entry of Zn^2+^ into mitochondria. The information summarised is derived from studies of TRPM2 in all animal cells. Not all intracellular locations and functions have been reported for hepatocytes or other cell types in liver. (References are given in the text.)

**Figure 3 antioxidants-10-01243-f003:**
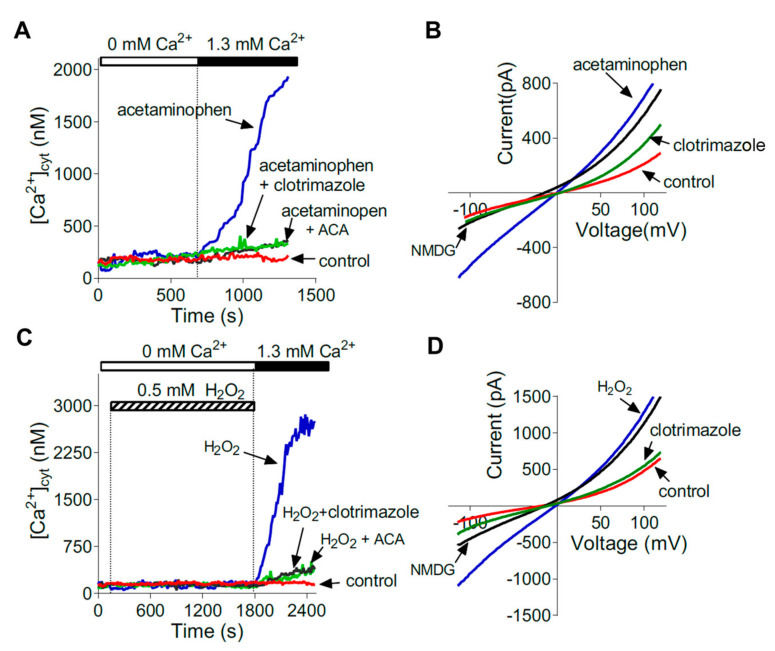
Evidence for the expression of functional TRPM2 channels in the plasma membranes of isolated mouse hepatocytes. TRPM2 channels were activated by acetaminophen (**A**,**B**) or H_2_O_2_ (**C**,**D**). Ca^2+^ entry was assessed by the increase in cytoplasmic Ca^2+^ concentration ([Ca^2+^]_cyt_) (**A**,**C**) or by patch clamp recording of TRPM2 currents (**B**,**D**). Properties of the currents activated by acetaminophen and H_2_O_2_ are similar to those observed for TRPM2 channels in other cell types. Evidence for TRPM2 channel activity is also provided by use of the TRPM2 inhibitors, clotrimazole, and N-(p-amylcinnamoyl) anthranilic acid (ACA). NMDG is N-methyl-d-glucamine. Reproduced with permission from Kheradpezhouh et al. [29].

**Figure 4 antioxidants-10-01243-f004:**
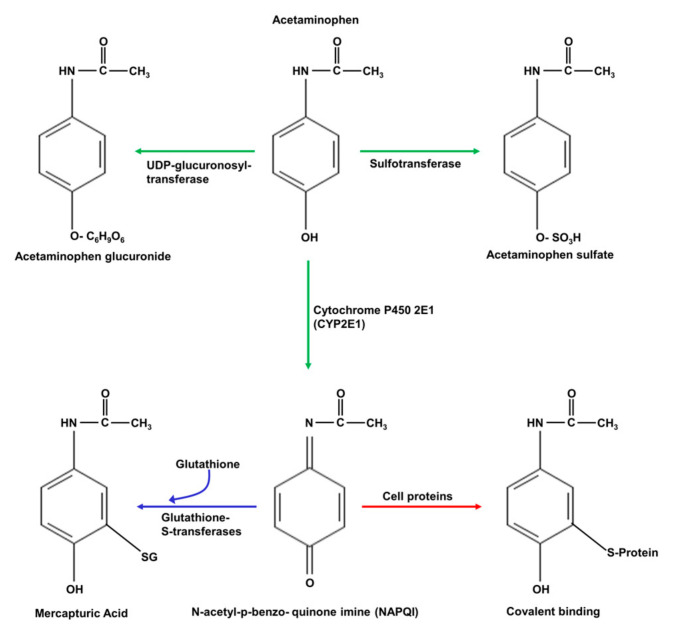
The pathways by which acetaminophen is metabolised in hepatocytes. Acetaminophen is initially metabolised by UDP-glucuronosyl transferase, sulfotransferase, and cytochrome P450 2E1 (green arrows). The resulting acetaminophen glucuronide and acetaminophen sulphate are excreted from hepatocytes via the bile canaliculus. In a reaction with glutathione (GSH), the product of cytochrome P450 2E1, N-acetyl-p-benzoquinone imine (NAPQI), is converted to mercapturic acid (blue arrow), which is also excreted in bile fluid. Acetaminophen can also be N-deacylated to form p-amino-phenol sulphate (not shown). At high doses of acetaminophen, such as in acetaminophen toxicity, the glutathione available for the conversion of NAPQI to mercapturic acid is exhausted. Consequently, NAPQI accumulates and reacts with, and inactivates, enzymes and other proteins in the liver (red arrow). Moreover, the deficiency in glutathione caused by high levels of NAPQI reduces the ability of hepatocytes to remove the ROS generated in other reactions, leading to further ROS-induced damage to proteins [64,68].

**Figure 5 antioxidants-10-01243-f005:**
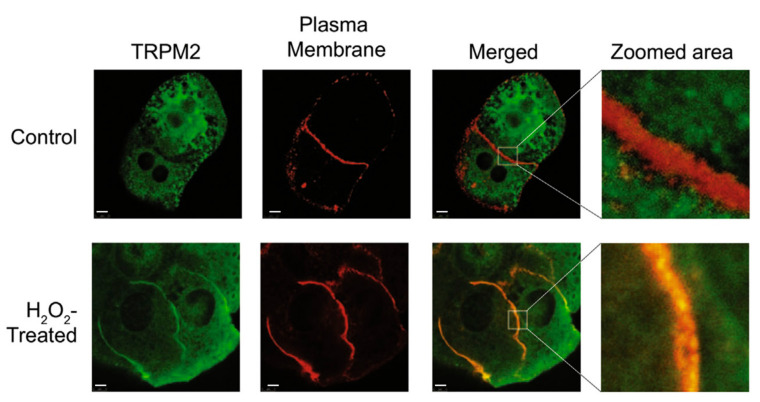
In rat hepatocytes, H_2_O_2_ induces the movement of some TRPM2 protein from intracellular locations to the plasma membrane. Confocal microscopy images (100× magnification) of untreated control hepatocytes (Control) and hepatocytes treated with H_2_O_2_ (H_2_O_2_ treated). Green fluorescence corresponds to TRPM2 and red fluorescence to cadherin (marker protein for the plasma membrane). Yellow indicates the overlap between green and red fluorescence (Merged and Zoomed area), representing TRPM2 and the plasma membrane, respectively. Reproduced with permission from [43], copyright 2018 Elsevier.

**Figure 6 antioxidants-10-01243-f006:**
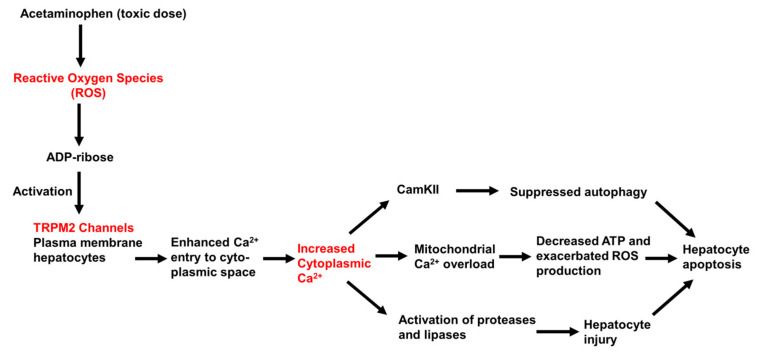
Schematic representation of the proposed role of TRPM2 in mediating liver toxicity caused by high (overdose) levels of acetaminophen. The increase in ROS, resulting from the metabolism of acetaminophen and generation of high levels of NAPQI (Scheme, Figure 5), are proposed to lead to the synthesis of ADP-ribose, activation of TRPM2, Ca^2+^ entry to the cytoplasmic space, and increased [Ca^2+^]_cyt_. This, in turn, is proposed to lead to mitochondrial Ca^2+^ overload, the activation of Ca^2+^-sensitive proteases and lipases, and the activation of Ca^2+^/calmodulin protein kinase II (CamKII) and supressed autophagy. The activation of these pathways eventually leads to cellular apoptosis and death. (References are in the text.)

**Figure 7 antioxidants-10-01243-f007:**
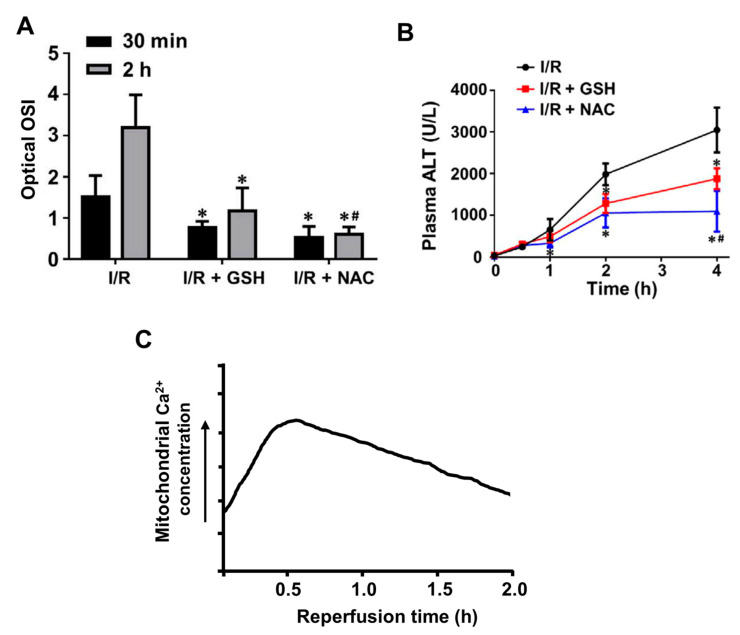
Increases in reactive oxygen species and intracellular Ca^2+^ in livers subject to ischemia and reperfusion. Comparison of changes in reactive oxygen species (oxidative stress) (**A**), blood concentrations of alanine transaminase, an indicator of liver damage (**B**), and mitochondrial calcium (**C**) following ischemia and reperfusion in mouse and rat models of liver ischemia–reperfusion injury. Oxidative stress is increased at 30 min and increases further at 120 min post initiation of the reperfusion (**A**). Mitochondrial calcium is increased at 5 min and reaches a peak at about 30 min post initiation of the reperfusion (**C**). Significant liver injury is apparent at 120 min post initiation of the reperfusion (**B**). (**A**) Oxidative stress index (OSI) measured in vivo at 30 min and 120 min following the beginning of the reperfusion in mouse livers subject to ischemia–reperfusion (I/R) without any pre-treatment, and following pre-treatment with the anti-oxidising agents glutathione or N-acetylcystamine (NAC). (**B**) Time courses for the increase in blood concentrations of alanine amino transferase following commencement of reperfusion for the livers shown in (**A**). (**C**) A schematic representation of the time course for the increase in mitochondrial calcium in mouse and rat livers subject to reperfusion following ischemia, based in the results of studies conducted by Isozaki et al. (2000), Pan et al. (2012), and Chattopadhyay et al. (2012) [78,81,82]. Oxidative stress index in (OSI in A) is defined as the ratio of the mean intensity of the sum of cellular H_2_O_2_ and HOCl (ROS) to the mean intensity of cellular glutathione (GSH). HOCl, H_2_O_2_, and glutathione were measured in vivo by two-photon dual imaging. * *p* < 0.05, compared with untreated groups; # *p* < 0.05, compared with I/R + GSH group. Panels (**A**,**B**) are reproduced from [77].

**Figure 8 antioxidants-10-01243-f008:**
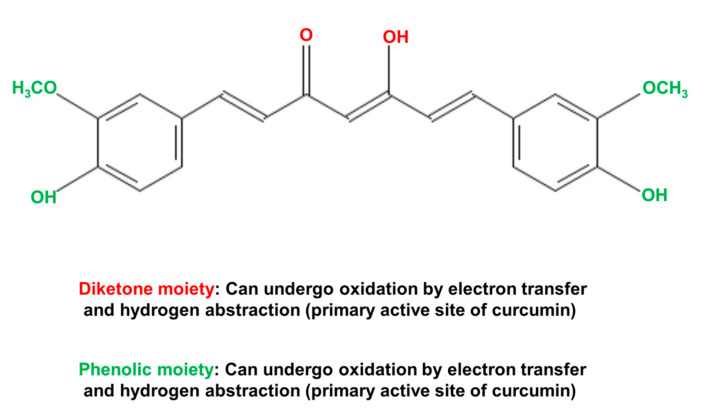
Chemical structure of the natural product curcumin (1E,6E)-1,7-bis(4-hydroxy-3-methoxyphenyl)-1,6-heptadiene-3,5-dione), which is derived from the spice turmeric (*Curcuma longa* Linn). The diketone moiety (shown in red) and the two phenolic moieties (shown in green) are the reactive groups responsible for many of the biological actions of curcumin [120].

**Figure 9 antioxidants-10-01243-f009:**
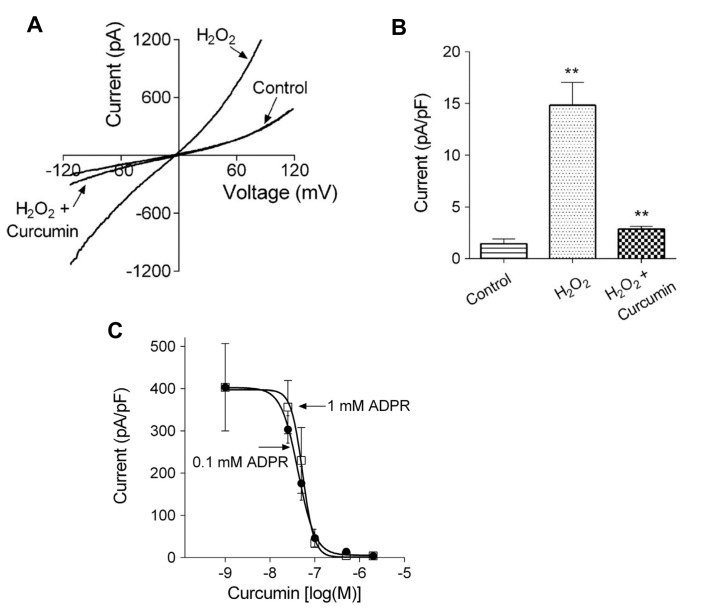
Curcumin inhibits TRPM2 non-selective cation channels, assayed by patch clamp recording, in isolated rat hepatocytes and in HEK293T cells heterologously expressing TRPM2. (**A**) Current–voltage plots for hepatocytes treated with H_2_O_2_ to activate TRPM2, measured in the presence and absence of 5 µM curcumin. (**B**) Average amplitude of the TRPM2-mediated Na^+^ current in untreated hepatocytes (Control), hepatocytes treated with H_2_O_2_ (H_2_O_2_) and in hepatocytes treated with H_2_O_2_ in the presence of 5 µM curcumin (H_2_O_2_ + Curcumin). ** *P* < 0.008 (**C**) Dose–response curves for the inhibition of the TRPM2 current activated by ADP-ribose (two concentrations administered via the patch pipette) to HEK293T cells heterologously expressing TRPM2. The IC_50_ values for curcumin inhibition are about 50 nM. Reproduced with permission from [119], copyright 2016 Elsevier.

**Table 1 antioxidants-10-01243-t001:** Inhibitors of TRPM2 channels.

General Classification of Inhibitor *	Name or Inhibitor (or Representative Example)		IC_50_ ** (µM)	Some Comments on Specificity	References
Repurposed inhibitors of other ion channels	*Gadolinium (*Gd^3+^)		0.27	Inhibits many classes of ion channels	[97,98,99]
	Flufenamic acid (FFA)		70	Inhibits numerous TRP channels, Ca^2+^-activated Cl^−^ channels, mitochondrial permeability transition pore	[96,100,101,102,103]
	2-(3-methylphenyl) aminobenzoic acid (3-MFA)		76	Inhibits numerous TRP channels	[102,104]
	Econazole		3–30	Inhibits numerous TRP channels, and store-operated Ca^2+^ entry channels	[100,101,104]
	Clotrimazole		3–30	Inhibits numerous TRP channels and Ca^2+^-activated potassium channels (KCa3.1), cytochrome P-450 and N-methyl D-aspartate receptors	[96,101,103,105]
	N-(p-amylcinnamoyl) anthranilic acid (ACA)		1.7	Inhibits numerous TRP channels and Ca^2+^-activated chloride channels	[96,101,103]
	(E)-2-(3-(3′-(Trifluoromethyl)-[1,1′-biphenyl]-4-yl)acrylamido)benzoic Acid (A23) (a derivative ACA)		0.78	Inhibits numerous TRP channels	[106]
	2-aminoethoxydiphenyl borate (2-APB)		1	Inhibits numerous TRP channels and store-operated Ca^2+^ entry channels	[96,107]
Analogues of endogenous TRPM2 ligands	Modifications of ADP-ribose: adenosine	8-Br-ADPR (8-bromo substitution)	300		[96,108]
		8-phenyl-ADPR (Adenosine modification with a bulky hydrophobic group)	11		[96,108,109]
		8-(3-thiophenyl)-ADPR (Adenosine modification with a bulky hydrophobic group)	51		[96,108,109]
		8-(3-Acetylphenyl)- ADPR (modification with a bulky hydrophobic group)	49		[96,108,109]
		8-Ph-2′-deoxy-ADPR (8-phenyl substitution of the adenine and 2′-deoxy modification at the adenosine ribose)	3		[96,108]
	Modifications of ADP-ribose: pyrophosphate	1″-*O*-Methyl-2″,2′,3″,3′-*O*-isopropylidene-5’-difluoromethylenediphosphoribose-2-chloroadenosine (7i)	5.7	Inhibits numerous TRP channels	[96,108,110]
		1″-*O*-Methyl-2″,3″-*O*-isopropylidene-5’-methylenediphosphoribose adenosine (8a)	5.4	Inhibits numerous TRP channels	[96,108,110]
	Modifications of ADP-ribose: terminal ribose	1″-deoxy-ADPR (Removal of hydroxyl group from terminal ribose)	ND		[96,110]
		3″-deoxy-ADPR (Removal of hydroxyl group from terminal ribose)	ND		[96,110]
		1″- β-O-Me-2″,3″-O-iPr-ADPR (Masking all hydroxyl groups on the terminal ribose)	ND		[96,110]
		α -1″-O-methyl-ADPR	ND		[96,111]
		THF-ADP	ND		[96,111]
		methyl-ADP	ND		[96,111]
		Adenosine monophosphate (AMP)	70		[112]
		8-Br-cyclic ADPR (a cyclic ADPR analogue)	100		[112]
	Peptide inhibitor	N-Tyr-Gly-Arg-Lys-Lys-Arg-Arg-Gln-Arg-Arg-Arg-Gly-Ser-Arg-Glu-Pro-Gly-Glu-Met-Leu-Pro-Arg-Lys-Leu-Lys-Arg-Val-Leu-Arg-Gln-Glu-Phe-Trp-Val-OH (tat-M2NX)	0.40	Designed to correspond to the C-terminal NUDT9-H domain	[96,103,113]
Molecules identified by high throughput screening	Scalaradial		0.21	Inhibits TRPM7 channels	[96,114]
	Ethyl (3-[4-(trifluoromethyl)phenyl]-5,6,7,8-tetrahydrocyclohepta[c]pyrazol-1(4H)-yl)acetate (JNJ-28583113)		0.013	One of the most potent TRPM2 inhibitors so far discovered. Membrane permeable but rapidly metabolised in vivo	[96,104]
Molecules identified by high virtual screening	2-(3-phenyl-1H-pyrazol-4-yl)-2,3-dihydroquinazolin-4(1H)-one (H1) (phenyl group at 3′-position of pyrazole ring)		3.7–5.1		[96,115]
	2,3-dihydro-quinazolin-4(1H)-one (D9) (It has naphthyl group at 3′-position of pyrazole ring, and C8 as Br)		3.7		[96,115]
Molecules identified by other approaches	2-Cyano-3-(3,4-dihydroxyphenyl)-*N*-(benzyl)-2-propenamide, 2-Cyano-3-(3,4-dihydroxyphenyl)-N-(phenylmethyl)-2-propenamide (Tyrphostin, AG490)		3.0	Inhibits TRPM2 activation within 2 min of pretreatment but does not affect intracellular Ca^2+^ level during opening state of TRPM2	[116]
	(*E*)-2-Cyano-3-(3,4-dihydroxyphenyl)-*N*-(3-phenylpropyl)-2-propenamide (AG555)		2.1	Stronger inhibitory effect on H_2_O_2_-induced TRPM2 activation than AG490	[117,118]
	(*E*)-2-Cyano-3-(3,4-dihydroxyphenyl)-*N*-(4-phenylbutyl)-2-propenamide (AG556)		0.94	Stronger inhibitory effect on H_2_O_2_-induced TRPM2 activation than AG490	[117,118]
	Curcumin ((1E,6E)-1,7-bis(4-hydroxy-3-methoxyphenyl)-1,6-heptadiene-3,5-dione)		0.05	Mechanism does not appear to involve direct block of the channel or a reduction in ROS	[119]

* Classification is based in that of Zhang et al. (2020) [96]. ADPR represents ADP-ribose. ** Most IC_50_ values have been obtained from dose–response curves determined by patch clamp recording of currents through channels with the characteristics of TRPM2 in the cell type under study, or were obtained from patch clamp recordings or fluorescence Ca^2+^ imaging in HEK293 cells heterologously expressing TRPM2. ND indicates “not determined” (inhibitor tested at two concentrations only).

**Table 2 antioxidants-10-01243-t002:** Biochemical and signalling pathways in animal cells known to be affected by curcumin ((1E,6E)-1,7-bis(4-hydroxy-3-methoxyphenyl)-1,6-heptadiene-3,5-dione), and some proteins which have been identified as targets, or putative targets, modified by curcumin.

Biochemical Pathway	Examples of Proteins That Have Been Identified as Targets, or Putative Targets, of Curcumin Action	References
Removal of reactive oxygen species *(Predominantly pathways activated)*	Catalase	[130,131]
	Superoxide dismutase	[130]
	Glutathione peroxidase	[130,131]
	Carbonyl reductase [NADPH]	[129,131]
	Peroxiredoxin-1	[129]
	Quinone oxidoreductase	[129,132]
	Glutathione S-transferase	[129]
	Heme oxygenase-1 (HO-1)	[133,134]
	Glutathione S-transferase	[129,135]
	NAD(P)H dehydrogenase, quinone 1	[129]
	NAD(P)H dehydrogenase, quinone 2	[129]
Cell proliferation pathways *(Predominantly pathways inhibited)*	Phospholipase C (PLC)	[136]
	Trophoblast cell surface antigen 2	[137]
	miRNA-7	[138]
	miRNA-let7c and miR-101 (via effects on EZH2 and NOTCH1)	[139]
Apoptotic pathways *(Predominantly pathways induced)*	Cleavage inhibitor of apoptosis (CIAP)	[136,140]
	Fas-associated protein with death domain (FADD)	[136,140]
	FLICE-like inhibitory protein (FLIP)	[136,140]
	JAK–STAT signalling pathway	[141]
	WNT signalling pathway	[141]
	Protein C-ets-1 (ETS-1)	[141]
	Death-inducing signalling complex (DISC)	[136]
	X-linked inhibitor of apoptosis protein (XIAP)	[136]
	Caspase 3/7	[142]
	Matrix metalloproteinase (MMP) (MMP-2, MMP-9)	[141,143,144]
Inflammation and cytokine pathways *(Predominantly pathways inhibited)*	Lipoxygenase (LOX)/cyclooxygenase (COX)	[133,134,141,145]
	Nuclear factor-κB (NF-κB)	[141,143,146]
	Tumour necrosis factor α	[131]
	Interleukin 1	[131]
	Interleukin 6	[131]
	Signal transducer and activator of transcription (STAT)	[141,147]
	Cyclin D1	[141,147]
	Inducible nitrogen oxide synthase (iNOS)	[148]
Protein kinase signalling pathways *(Activation or inhibition depends on cell conditions (normal* vs. *injured/transformed) and types))*	Focal adhesion kinase (FAK)	[149]
	Protein kinase (PK) C	[150]
	Phosphatidylinositol-3-kinase (PI3K)	[143]
	Mechanistic target or rapamycin (mTOR)	[140,143]
	Cyclin-dependent kinase (CDK)	[143]
	Mitogen activated protein kinase	[140]
	Inhibitor of κB kinase	[140]
	Protein kinase B (Akt)	[131]
Growth factors signalling pathways *(Activation or inhibition depends on cell conditions (normal* vs. *injured/transformed) and types))*	Human epidermal growth factor receptor 2	[143]
	Vascular endothelial growth factor receptor (VEGFR)	[140,143]
	Vascular endothelial growth factor (VEGF)	[140,143]
	Fibroblast growth factor (FGF)	[136,140]
	Epidermal growth factor (EGF)	[136,140,151]
	Platelet-derived growth factor (PDGF)	[136]
Ion channels and transporter signalling and biochemical pathways *(Activation or inhibition depends on cell conditions (normal* vs. *injured/transformed) and types))*	High voltage-gated Ca^2+^ channels	[152]
	Cav2.2 and Cav2.1	[152,153]
	Ca^2+^ release-activated Ca^2+^ (CRAC) channels	[152,153]
	Cystic fibrosis transmembrane conductance regulator (CFTR)	[152]
	Aquaporin 4 (AQP-4)	[152]
	Voltage-Gated Potassium Channel Kv1.3 (Kv1.3)	[152,154]
	ATP-binding cassette drug transporter	[152]
	Glucose transporter protein 2 GLUT2	[152]
	Glucose transporter protein 4 (GLUT4)	[152]
	Transient receptor potential cation channel, subfamily M, member 2 (TRPM2)	[119]

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
