# Peer review of "TRPM2 Non-Selective Cation Channels in Liver Injury Mediated by Reactive Oxygen Species"

_antioxidants, 2021, doi:10.3390/antiox10081243_

Round 1

Reviewer 1 Report

The authors summarise the current knowledge of the roles of TRPM2 channels and intracellular calcium ions in mediating ROS-initiated liver injury and the progression of liver disease and liver injury to end-stage liver disease or liver failure. The review is very comprehensive but a few aspects need to be tackled. In particular, the figure indicating the expression of TRPM2 in hepatocytes and Kupffer cells in paraffin sections of the human liver detected by in situ hybridization (ISH) is not clear and misleading. I do understand that this image comes from Badr et al. published in another journal, but it is not convincing as an ISH expert. The authors need to better explain this image clarifying in the legend more details or delete it. Moreover, discordant data are present in the literature associated with two molecules. These aspects need to be clarified. 2-Cyano-3-(3,4-dihydroxyphenyl)-N-(benzyl)-2-propenamide, 2-Cyano3-(3,4-dihydroxyphenyl)-N-(phenylmethyl)-2-propenamide (Tyrphostin, AG490)
does not always fully blocks H2O2-induced increase in intracellular Ca2+ in
TRPM2. Please clarify these details. Much more complex are the interactions of curcumin ((1E,6E)-1,7-bis(4-hydroxy-3-methoxyphenyl)-1,6-
heptadiene-3,5-dione) in my opinion and it cannot easily be indicated that it is "thought to inhibit a step in the pathway by which ADPR activates TRPM2"

Author Response

  1. In particular, the figure indicating the expression of TRPM2 in hepatocytes and Kupffer cells in paraffin sections of the human liver detected by in situ hybridization (ISH) is not clear and misleading. I do understand that this image comes from Badr et al. published in another journal, but it is not convincing as an ISH expert. The authors need to better explain this image clarifying in the legend more details or delete it.

We have deleted the in situ hybridization figure. (We have been concerned about the quality of this figure all along. But wanted initially to include the figure as it could make the Review more interesting (not just words). We contacted Dr Mori corresponding author of the original paper and he kindly sent a high-resolution image. However, in the end we agree that the comment of the reviewer that this is not up to standard for in situ hybridization is paramount.)

All subsequent figures have been re numbered.

2.Moreover, discordant data are present in the literature associated with two molecules. These aspects need to be clarified. 2-Cyano-3-(3,4-dihydroxyphenyl)-N-(benzyl)-2-propenamide, 2-Cyano3-(3,4-dihydroxyphenyl)-N-(phenylmethyl)-2-propenamide (Tyrphostin, AG490)
does not always fully blocks H2O2-induced increase in intracellular Ca2+ in
TRPM2. Please clarify these details.

We have amended the comment in Table 1 relating to this TRPM2 inhibitor.

3.Much more complex are the interactions of curcumin ((1E,6E)-1,7-bis(4-hydroxy-3-methoxyphenyl)-1,6-
heptadiene-3,5-dione) in my opinion and it cannot easily be indicated that it is "thought to inhibit a step in the pathway by which ADPR activates TRPM2"

We have amended the comment in Table 2 related to curcumin. This is now in line better with the comment in the text further below the Table in proposed mechanism if curcumin inhibition. I realise that our comment in Table 2 overstepped the hypothesis/ proposal in the original paper.

Reviewer 2 Report

Eunus S. Ali and colleagues reviewed TRPM2 and related ROS-injury in hepatocytes. This review covered a wild range of roles of TRPM2.

I list some minor points:

L145 Authors should show the structure of TRPM2. I could not imagine the protein.

L161 ADR-ribose should be ADP-ribose

L163 Authors should show how PIP2 activates TRPM2. (Should be “The activation”)

Figure 2 Authors should indicate poly ADP-ribose in Fig.2

L177 How does PARP produce ADPR?

Figure 4 / 6 / 8 should replace with the high-resolution fig.

L285 Concentration of acetaminophen should be provided; physiological levels?

L495 Remove a period. Figure. 8

Author Response

Response to Reviewers Comments Reviewer 2

L145 Authors should show the structure of TRPM2. I could not imagine the protein.

In Fig. 2, we have added “S1” label to the scheme of TRPM2 polypeptide in the plasma membrane (top RHS Fig. 2. This is the structure of TRPM2.) In the text we have specifically referred the reader to this part Fig. 2 ie the structure TRPM2 scheme. Lines 141-144. (We considered adding a separate panel to Fig. 2, but believe the above does address and clarify this issue.)

L161 ADR-ribose should be ADP-ribose

Now corrected, L171

L163 Authors should show how PIP2 activates TRPM2. (Should be “The activation”)

Text modified to include more information on PIP2 mechanism. L173-175 legend Fig 2. L187-199 text, plus new reference 38

Figure 2 Authors should indicate poly ADP-ribose in Fig.2

Fig 2 modified to include polyADP-ribose in nucleus, and legend modified.

L177 How does PARP produce ADPR?

We think this is addresses now with the above pathway indicated better in the nucleus of Fig. 2 scheme.

Figure 4 / 6 / 8 should replace with the high-resolution fig.

We will upload high resolution TIF files with revised Ms.

L285 Concentration of acetaminophen should be provided; physiological levels? 

We have added some sentences to indicate approximate dose levels for normal and toxic acetaminophen plus adding in about 3 new references to support these values. L310-316

L495 Remove a period. Figure. 8.

Have removed period.